# Feasibility of Rehabilitation during Chemoradiotherapy among Patients with Stage III Non-Small Cell Lung Cancer: A Proof-of-Concept Study

**DOI:** 10.3390/cancers14102387

**Published:** 2022-05-12

**Authors:** Melissa J. J. Voorn, Bart C. Bongers, Vivian E. M. van Kampen-van den Boogaart, Elisabeth J. M. Driessen, Maryska L. G. Janssen-Heijnen

**Affiliations:** 1Department of Clinical Epidemiology, VieCuri Medical Centre, 5912 BL Venlo, The Netherlands; li.driessen@zuyderland.nl (E.J.M.D.); mjanssenheijnen@viecuri.nl (M.L.G.J.-H.); 2Adelante Rehabilitation Centre, 5912 BL Venlo, The Netherlands; 3Department of Epidemiology, GROW School for Oncology and Developmental Biology, Faculty of Health, Medicine and Life Sciences, Maastricht University, 6200 MD Maastricht, The Netherlands; 4Department of Nutrition and Movement Sciences, (NUTRIM) School of Nutrition and Translational Research in Metabolism, Faculty of Health, Medicine and Life Sciences, Maastricht University, 6200 MD Maastricht, The Netherlands; bart.bongers@maastrichtuniversity.nl; 5Department of Epidemiology, Care and Public Health Research Institute (CAPHRI), Faculty of Health, Medicine and Life Sciences, Maastricht University, 6200 MD Maastricht, The Netherlands; 6Department of Pulmonology, VieCuri Medical Centre, 6200 MD Venlo, The Netherlands; vvankampen@viecuri.nl

**Keywords:** lung cancer, chemoradiotherapy, home-based rehabilitation, physical exercise training, feasibility, adherence, patient experiences and preferences

## Abstract

**Simple Summary:**

Currently, patients with a poor physical status undergoing chemoradiotherapy (CHRT) for stage III non-small cell lung cancer (NSCLC) are at high risk for poor treatment tolerance and poor survival. Rehabilitation during CHRT might prevent an often-observed reduction in physical fitness and deterioration in nutritional status. In this proof-of-concept study, we investigated whether a multimodal program for rehabilitation during CHRT, constructed in collaboration with patients and healthcare professionals, was feasible with respect to program adherence during CHRT, patient motivation, patient preferences and experiences, dropout rate, adverse events, and logistic planning. Results demonstrate that supervised and personalized rehabilitation in patients with stage III NSCLC undergoing CHRT seems feasible when the intensity of the physical exercise training program and nutritional advice are adjusted to the possibilities and preferences of the patients. It is recommended to design a supervised and personalized rehabilitation program with a low-to-moderate training intensity and a longer training session duration.

**Abstract:**

Rehabilitation during chemoradiotherapy (CHRT) might (partly) prevent reduction in physical fitness and nutritional status and could improve treatment tolerance in patients with stage III non-small cell lung cancer (NSCLC). The aim of this proof-of-concept study was to investigate the feasibility of a multimodal program for rehabilitation during CHRT. A home-based multimodal rehabilitation program (partly supervised moderate-intensity physical exercise training and nutritional support) during CHRT was developed in collaboration with patients with stage III NSCLC and specialized healthcare professionals. A predetermined number of six patients with stage III NSCLC (aged > 50 years) who underwent CHRT and participated in this program were monitored in detail to assess its feasibility for further development and optimization of the program. The patient’s level of physical functioning (e.g., cardiopulmonary exercise test, six-minute walking test, handgrip strength, body mass index, fat free mass index, energy and protein intake) was evaluated in order to provide personalized advice regarding physical exercise training and nutrition. The program appeared feasible and well-tolerated. All six included patients managed to perform the assessments. Exercise session adherence was high in five patients and low in one patient. The performed exercise intensity was lower than prescribed for all patients. Patients were motivated to complete the home-based rehabilitation program during CHRT. Preliminary effects on physical and nutritional parameters revealed relatively stable values throughout CHRT, with inter-individual variation. Supervised and personalized rehabilitation in patients with stage III NSCLC undergoing CHRT seems feasible when the intensity of the physical exercise training was adjusted to the possibilities and preferences of the patients. Future research should investigate the feasibility of a supervised and personalized rehabilitation program during CHRT with a low-to-moderate exercise intensity with the aim to prevent physical decline during CHRT.

## 1. Introduction

Lung cancer is the leading cause of cancer-related death all over the world [1], accounting for 18.4% of all cancer-related deaths in 2018 [2]. Approximately 85–90% of the patients with lung cancer suffer from non-small cell lung cancer (NSCLC) [1]. Stage III disease constitutes around 20% of all NSCLC cases [3]. Standard treatment for patients with stage III NSCLC is concurrent or sequential chemoradiotherapy (CHRT) [4]. Patients with stage III NSCLC often have characteristics that increase the risk for treatment complications, such as a higher age (≥50% is aged over 70 years), smoking-related comorbidity, and poor physical performance status [5,6,7,8]. Furthermore, frailty, long-term physical inactivity, and malnutrition are often present [9], which can decrease treatment tolerance, survival, and quality of life [6,10]. Rehabilitation during CHRT might be effective to improve treatment tolerance, quality of life, and survival during intensive treatment with CHRT [11,12,13].

Although patients with lung cancer perceive physical activity as being important for recovery during and after treatment, most patients are insufficiently active. Previous studies in patients with NSCLC show that the willingness and ability to participate in a prehabilitation program (between 28% and 56% [14,15]) is low and that program adherence is only moderate (between 53% and 73% [14,16]). Among dropout reasons, cancer-related side effects and, mostly, lack of interest and motivation represent key contributors [17]. Nevertheless, high adherence of patients to rehabilitation during CHRT is crucial to reduce treatment complications [18]. Understanding what amount of training volume is feasible, thereby including patient preferences, is important to ensure that rehabilitation during CHRT is personalized and that patients and their informal caregivers are both able and willing to participate and adhere to the program.

Studies including patients with lung cancer have shown that intramural or extramural physical exercise before, during, and after treatment might counteract adherence because of commuting problems, accessibility of services, comorbidity, and vulnerability [11,19]. Self-monitoring might increase motivation for exercise continuation, as insights regarding improvements can be gained quickly (with a pedometer or with low complexity walk and stair climbing tests), as has been shown in patients with rectal cancer [19,20]. Furthermore, peer support and the involvement of close relatives increase adherence rates in patients who undergo surgery [21,22]. A personalized supervised program that is home-based and takes into account the preferences and experiences of patients might improve motivation and adherence in patients with stage III NSCLC undergoing CHRT.

Unfortunately, evidence on the feasibility of multimodal rehabilitation during CHRT among patients with stage III NSCLC is lacking. Therefore, the aim of this study was to investigate whether a multimodal program for rehabilitation during CHRT, constructed in collaboration with patients and healthcare professionals, was feasible with respect to adherence to the rehabilitation program during CHRT, motivation, patients’ preferences and experiences, dropout rate, adverse events, and logistic planning.

## 2. Methods

### 2.1. Patients

In this single-group prospective proof-of-concept study, a predefined number of six patients participated. Patients were checked for eligibility by their treating pulmonologist at VieCuri Medical Centre between February 2019 and March 2021. These patients were diagnosed with stage III NSCLC according to the 8th edition of the TNM guidelines and were referred for and underwent CHRT (either concurrent CHRT or sequential CHRT). Patients were eligible when they were aged ≥50 years and provided written informed consent. Patients unable to perform a moderate-intensity physical exercise program, with a diagnosis of previous cancer in the past three years, or psychological or somatic constraints that might limit their ability to cooperate with study procedures were excluded. The Medical Research Ethics Committee of Maastricht UMC + decided that this study met the ethical policies and regulations of the Dutch government (non-WMO statement 2017-0154).

### 2.2. Cancer Treatment

Patients enrolled in this study received standard treatment including at least thirteen weeks of concurrent CHRT or eighteen weeks of sequential CHRT. Regimens for chemotherapy consisted of two or three concurrent cisplatin or carboplatin-based doublet cycles or three to four sequential cisplatin or gemcitabine doublet cycles. Radiotherapy was delivered with an arc technique and delivered using 6–10 MV photons. Gross tumour volume included the primary tumour and pathologic lymph nodes as identified on the fluorodeoxyglucose-positron emission tomography scan. Volume constraints for the oesophagus have not been performed. The maximum point dose in the oesophagus is 76 Gy (biologically equivalent dose in 2 Gy fractions). The clinical target volume had an extra margin to include regions at risk of microscopic extension. Planning target volume encompassed a margin for inter- and intrafraction patient and organ motion. Schedules were 33 × 2 Gy (once daily), 24 × 2.75 Gy (once daily), and radiotherapy according to an individualized prescribed maximal tolerated dose protocol (once or twice daily).

### 2.3. Content and Assessment of the Multimodal Rehabilitation Program during CHRT

After informed consent was obtained, baseline data, including demographics sex, age, World Health Organization (WHO) performance status, and Charlson comorbidity index [23], were collected from the electronic patient file. The schedule of enrolment, interventions, and assessments for patients who underwent rehabilitation during CHRT is shown in Table 1.

The aim of the pre-treatment baseline assessment (T0) was to evaluate the patient’s level of physical functioning (e.g., physical functioning parameters, nutritional parameters) in order to provide personalized advice regarding physical exercise training and nutrition. To monitor changes and subsequently adjust the program, baseline assessments, excluding the cardiopulmonary exercise test (CPET), were repeated between the first and second course of chemotherapy (T1) and after the last session of radiotherapy (T2). Three months after treatment (T3), all assessments, including the CPET, were repeated again. Preliminary effects of the program were evaluated by changes in physical and nutritional parameters during assessments T0, T1, T2, and T3. A blueprint of the rehabilitation program during CHRT was developed and discussed in the multimodal and transmural project team, as well as with patients (representatives). The rehabilitation program during CHRT, which was incorporated into the patient’s cancer treatment schedule, consisted of physical exercise training, a nutritional support module, and smoking cessation. The length of rehabilitation during CHRT depended on the duration of treatment, including treatment delay.

#### 2.3.1. Physical Exercise Training

The blueprint of the physical exercise training program was developed according to the International Consensus on Therapeutic Exercise and Training (i-CONTENT) scale and is presented in Table 2.

Emphasis was put on the preferences of the patient by personalization of functional exercises that were of relevance and meaningful to a patient. The thirteen- or eighteen-week physical exercise training program was carried out in the patient’s living environment. Once every two weeks, a training session was supervised by a physical therapist specialized in oncology. Training frequency was five times a week and started at a duration of at least fifteen minutes, which was progressively increased to 45 min. Training intensity for the home-based sessions was tailored using the 6–20 Borg scale for rating of perceived exertion, aiming at a moderate intensity with a Borg score of 13–15 [24]. Aerobic training consisted of a patient’s preferred activities involving large muscle groups (e.g., walking, cycling, climbing stairs, swimming). Peripheral resistance exercises of the large muscle groups of the lower and upper extremities using functional open and closed kinetic chain exercises (e.g., stair climbing, sit-to-stand exercises, a Thera band, filled 0.5-L bottles) were designed for each individual according to the relevant training zones (6–20 Borg score of 13–15). For inspiratory muscle training (IMT), patients performed two daily sessions of 30 breaths using an inspiratory muscle trainer (Threshold IMT, Philips Respironics, Murrysville, PA, USA) at the highest tolerable intensity [25]. The initial resistance (cm H_2_O) was set and increased using the 6–20 Borg scale [13,14,15], which was also based on the patient’s symptoms of fatigue, dyspnoea, or pain.

#### 2.3.2. Nutritional Support

Nutritional counselling was performed based on the standard protocol for patients with cancer in the general [26] and elderly population [27]. Individualized counselling aimed to educate patients on how to modify their usual meals by making them adhere to individual energy, protein, and other macronutrient requirements. The dietary advice aimed to specify the type and amount of food, the number of meals, and calorie or protein amounts to achieve daily or dietary recommendations as part of standard care. Advice by a dietician was personalized to a patient’s eating pattern and preferences.

#### 2.3.3. Smoking Cessation

Patients who smoked were encouraged to quit. When applicable, patients were referred to their general practitioner for a smoking-cessation program (according to Dutch guidelines) [28]. During contact moments with the healthcare professionals, patients were motivated to persevere with smoking-cessation.

#### 2.3.4. Physical Assessments

The pulmonologist referred patients to the sports physician for a CPET to examine the patient’s aerobic fitness and provided personalized advice regarding the content of the physical exercise program. An incremental CPET was performed on an electronically braked cycle ergometer (Monark LC6 novo, Exercise AB, Vansbro, Sweden) according to ATS/ERS standards [29] with respiratory gas analysis measurements. CPET interpretation was performed by a sports physician. Absolute oxygen uptake (VO_2_) at peak exercise (VO_2peak_) was calculated as the average value over the last 30 s prior to termination of the test. Peak heart rate was defined as the highest heart rate achieved during the CPET [30]. In order to personalize rehabilitation during CHRT, the physical therapist aimed to evaluate the level of physical functioning using performance-based tests to estimate functional walking distance (six-minute walk test (6MWT)), muscle strength (handgrip strength (HGS)), and daily physical activity level (pedometer). Functional walking distance was assessed by the 6MWT according to the ATS guideline [31]. After the test, patients were asked to rate their individually perceived exertion using the 6–20 Borg scale for rating of perceived exertion [24]. Maximal handgrip strength was assessed with a Jamar hydraulic hand dynamometer (J.A. Preston Corporation, Jackson, MI, USA). The highest value of three attempts for the dominant hand was registered [32]. Daily physical activity level was assessed by a pedometer (HiTrax Walk pedometer, TFA Dostmann, Wertheim, Germany).

#### 2.3.5. Nutritional Assessment

In order to provide tailored nutritional advice, the dietician aimed to evaluate a patient’s nutritional status with the body mass index (BMI), fat free mass index (FFMI), and energy requirement. In addition, the dietician provided insight into the protein requirement, which was estimated using the formula of Gallagher [33]. Body height was measured in standing position without shoes. Body mass was measured without shoes and coat. Body composition assessment (e.g., FFMI) was performed by direct segmental multi-frequency bioelectrical impedance analysis (Seca Medical Body Composition Analyzer 515, Hamburg, Germany). The estimates obtained are from the manufacturer’s proprietary algorithms with patients standing.

### 2.4. Feasibility of the Performed Assessments and the Rehabilitation Program during CHRT

The feasibility of performing the assessments and the rehabilitation program during CHRT was measured by adherence to the rehabilitation program during CHRT, motivation, patient preferences and experiences, patient dropout and adverse events during rehabilitation, as well as by logistical planning of the inclusion of patients and communication in this multimodal setting. Adherence to the rehabilitation program was monitored with an exercise diary and weekly feedback from the patients. A high exercise session adherence was defined as achieving ≥80% of the prescribed training session frequency and duration throughout the training program. Motivation was measured after each supervised physical exercise training session by asking patients to rate their motivation of performing the rehabilitation program, with help of a visible analogue scale (VAS) from 0–10, in which 10 meant excellent motivation. Involvement of an (in)formal caregiver was encouraged to promote motivation. Patient preferences and experiences (e.g., beneficial effects, deficiencies, impediments, (transmural) logistical problems) regarding the performance of physical and nutritional assessments, physical exercises, nutritional approach, willingness to quit smoking, supervision, and social support were recorded via usual care appointments with the healthcare professionals every two to three weeks. Patient dropout and adverse events from treatment or rehabilitation were collected by the healthcare professionals during contact moments as part of usual care. Logistical planning was discussed every six months with healthcare professionals from the three involved healthcare organizations (hospital, radiation centre, and rehabilitation centre within the hospital).

### 2.5. Statistical Analyses

Data were analysed by using IBM SPSS Statistics version 24.0. Detailed descriptive analyses were performed to describe the feasibility of the rehabilitation program during CHRT at each time point. Statistical significance was not determined as the included patient group was too small for any kind of valid statistical testing.

## 3. Results

### 3.1. Patient and Treatment Characteristics

Seven patients were invited to participate in this proof-of-concept study. One patient refused participation, because of being vulnerable and having a very limited social network. Six patients completed the rehabilitation program during CHRT. Patient characteristics, treatment schedule, physical and nutritional parameters, and the feasibility of rehabilitation during CHRT are presented in Table 3 and Table 4. All patients in this study started chemotherapy within four days after baseline assessments. Four patients received concurrent CHRT and two patients received sequential CHRT. The chemotherapy regimens for NSCLC patients were docetaxel, vinorelbine, and pemetrexed in one patient, and cisplatin + gemcitabine in five patients. Radiotherapy was followed as prescribed (33 × 2 Gy or 24 × 2.75 Gy) once daily in all patients. Adverse events that were judged to be related to chemotherapy were anaemia (two patients, 33%) and hypoalbuminemia (four patients, 67%).

### 3.2. The Multimodal Rehabilitation Program during CHRT

To improve aerobic fitness, five patients chose to walk, and one patient choose to cycle and swim at least five times a week. All patients lived in a single-family home. As such, stairs, a chair, bench, and table could be used to perform resistance exercises. All patients received nutritional advice that supported physical exercise training by ensuring sufficient protein intake. All patients reported a history of smoking and current smokers (n = 4) were advised to quit smoking with the guidance of the general practitioner. Two patients had stopped smoking two years before the diagnosis of lung cancer.

### 3.3. Feasibility

#### 3.3.1. Feasibility of the Rehabilitation Program during CHRT

Outcomes of patient characteristics, physical and nutrition assessments, and adherence to the multimodal rehabilitation program during CHRT are shown in Table 3 and Table 4. Adherence to the rehabilitation program was high in five patients as they completed ≥80% of the prescribed sessions and increased session time of aerobic and resistance exercises from 15 to 45 min. Adherence was low (48%) in one patient due to fatigue and decreased mood. None of the patients were able to increase the resistance of the IMT during the cancer treatment. Five patients were able to maintain the same IMT resistance and to maintain the IMT daily exercise frequency. One patient failed to exercise with the IMT device due to anxiety and shortness of breath. Alternative resistance exercises for the respiratory muscles (e.g., shoulder press, lateral raises, dumbbell press) were offered to this patient. All patients completed the exercise diary adequately. However, merely three patients reported occasionally the daily number of steps taken with the pedometer. Patients who did not adequately fill out the exercise diary reported that they forgot this and found it difficult to keep thinking about it, especially due to the long period of CHRT. All patients performed functional, resistance, and IMT training with a Borg score of 11–12 as perceived training intensity. Hence, none of the patients achieved the advised training intensity of 13–15 on the 6–20 Borg scale during these exercises. Four patients had mild swallowing irritation of the trachea during radiotherapy. These patients received adapted nutritional advice to improve the safety and comfort of eating, aiming to maintain an adequate nutritional intake. The intensity of physical exercise training had to be adjusted due to side effects of CHRT, such as fatigue, shortness of breath, and pain in all patients. There were no dropouts or adverse events as a result of the rehabilitation program. Regarding patient preferences and experiences, patients experienced their cancer treatment as an intense period in which many appointments took place, especially during radiotherapy. Patients reported feeling comfortable and safe because of the short lines between healthcare professionals as there was direct coordination between them when questions or uncertainties were posed by the patient. It was notable that patients needed and preferred intensive coaching during the first two weeks, and patients indicated that processing the situation (e.g., diagnosis, treatment, being concerned about their future) was stressful. Three patients (50%) indicated that the physical exercise training could be executed well, also during CHRT. One of the main reasons for being able to persevere was that the physical exercises and daily activities were performed together with their informal caregiver. Patients liked to perform the physical exercise training sessions at home. Regarding motivation, all patients indicated that they were motivated to participate in the study. This motivation remained relatively stable throughout the rehabilitation program. Patients indicated that it was difficult to remain motivated to perform the exercises during the weeks of chemotherapy and radiotherapy due to the many hospital appointments for the chemotherapy and radiotherapy and fatigue. Patients reported that they were better able to adhere to the interventions in the presence of healthcare professionals, but patients had difficulties motivating themselves to adhere to the frequency and intensity of the program during the unsupervised sessions. In five patients (83%), informal caregivers were actively involved. These patients indicated that their informal caregivers were a major source of social support and motivated them to continue the rehabilitation program. The patient with low adherence (48%) did not have an involved caregiver during rehabilitation.

There were problems concerning logistical planning in the multimodal setting. The inclusion of patients took a long time (25 months). However, six of the seven patients who were eligible and asked to participate in the study period immediately agreed to participate. The main reasons for the long inclusion period were a lack of attention for the study by the referring pulmonologists, the high work pressure at the outpatient clinic, and the COVID-19 pandemic. Communication between case managers, pulmonologists, and healthcare professionals was good. However, it was experienced that it would be difficult to keep each other informed about changes in treatment programs. Working at different healthcare organizations (hospital, rehabilitation centre, and radiation clinic) with different electronic patient files was experienced as a barrier by healthcare professionals. The time between the diagnosis and inclusion of patients was short, partly due to additional diagnostic investigations. The consequence of the multiple healthcare organizations involved was a complexity in patient referrals, and therefore patient inclusion for this study was difficult. In addition, inclusion was made more difficult by the impact of the COVID-19 pandemic.

#### 3.3.2. Feasibility of Performing Assessments

One patient was unable to perform a part of the T2 assessments because of the lockdown due to the COVID-19 pandemic. Two patients experienced the CPET as very intense and were reluctant to perform it on forehand. However, they were able to complete the CPETs. Patients were able to visit the hospital during all assessments. One patient had completed the 6MWT with a Borg score of 13–15. The other patients completed the 6MWT with a Borg score ≤12. The physical therapist observed that these patients were not short of breath or tired, but the patients indicated that they had walked as many meters as possible. The other physical and nutritional parameters were performed by all patients at all assessments.

### 3.4. Preliminary Effects on Physical and Nutritional Parameters

Preliminary changes in physical and nutritional assessments are shown in Table 3 and Table 4 and Figure 1.

#### 3.4.1. Physical Parameters

Regarding aerobic fitness, three patients (50%) deteriorated in VO_2peak_ between T0 and T3 and one patient (17%) improved between T0 and. In one patient, a 19% decrease in VO_2peak_ could be partly explained by a significant weight gain (62.9 kg at T0 versus 72.9 kg at T3). Mean VO_2_ at the VAT at T0 was 11.1 mL/kg/min and at T3 10.5 mL/kg/min). Distance on the 6MWT between T0 and T3 had improved with a mean percentage of 4.3%. HGS remained stable between T0 and T3.

#### 3.4.2. Nutritional Parameters

BMI and FFMI increased slightly in all patients during treatment. However, FFMI reduced in one patient between T0 and T3. All patients improved their protein and energy intake during treatment (T1 and T2) with a small decrease at T3. In three patients (50%), protein intake at T0 was too low (<80% of required). In two patients (33%), energy intake was too low at T0 (<80% of required). In three patients (33%), energy and protein intake improved after the first consultation and remained relatively stable during treatment. In the other three patients (33%), energy and protein intake decreased again at T3.

## 4. Discussion

This proof-of-concept study aimed to investigate whether multimodal rehabilitation during CHRT was feasible in patients with stage III NSCLC. Patients showed good training session adherence and adhered to the nutritional advice during multimodal rehabilitation. However, patients were unable to adhere to the prescribed training intensity. Although the supervision of healthcare professionals and the involvement of an informal caregiver led to better motivation, adherence to physical exercise training and dietary advice during CHRT was reported to be challenging as a consequence of fatigue and decreased mood. At group level, physical and nutritional parameters remained relatively stable during CHRT. However, large variety existed in response to rehabilitation, as some patients showed large improvements in physical and nutritional outcome measures, whereas others showed no progression or even deteriorated.

Patients indicated having difficulties in adhering to rehabilitation during the intensive treatment with CHRT, which might be due to the fact that patients with NSCLC often suffer from smoking-related comorbidities, physical inactivity, and frailty [5,6], making adherence particularly challenging.

The proposed physical exercise training intervention in the current study aimed for a moderate training intensity (Borg score 13–15). Exercise session adherence was high (≥80%) in five patients with the prescribed session time, and low (48%) in one patient. Adhering to moderate intensity exercises was not feasible for the patients. In a guideline from the American Cancer Society [33], it is reported that patients receiving chemotherapy and/or radiation therapy may need to exercise at a lower intensity and/or for a shorter duration during their treatment to maintain strength, which might help to counteract fatigue and depression. It has been hypothesized that home-based low-intensity physical exercise training programs may be easier for patients to complete during chemotherapy [34], whereas higher intensity, supervised exercise programs that incorporate resistance training and aerobic exercise may be most effective to improve physical fitness [35]. It could be questioned whether training at these low intensities provides sufficient overload to improve physical fitness. However, the aim of rehabilitation during CHRT should not be to improve, but to preserve physical fitness.

Although no randomized clinical trial on rehabilitation during CHRT in patients with NSCLC has yet been conducted, it was hypothesized for this study that rehabilitation during CHRT prevents the expected decline in physical fitness and reduces treatment complications. In a previous study in patients with NSCLC [36], lower physical activity levels during chemotherapy and radiotherapy were associated with complications during and after treatment. Preliminary results of the current study showed no noticeable decline in physical activity levels. The absence of decline can be perceived as a gain for this group who would otherwise have been expected to deteriorate [36]. In studies including patients with rectal cancer [20,37], supervised physical exercise training during CHRT has also demonstrated promising results to minimize physical decline and prevent the often observed decline in physical fitness during chemoradiotherapy.

In this proof-of-concept study, patients reported positive experiences with the support and guidance of the multidisciplinary team. Patients felt that it was easy to adhere to the prescribed exercises and nutritional advice, because they were home-based and personalized to their preferences and tailored to individual needs. The perceived importance of personalised interventions is in accordance with a previous retrospective study [38], in which patients indicated that a walking intervention after treatment for lung cancer was accessible, as walking was experienced as a familiar and enjoyable form of exercise and was therefore easy to adhere to. Furthermore, another benefit of home-based exercise is that it potentially increases long-term adoption and maintenance of physical activity as part of the patient’s daily routine [39,40]. In the current study, patients particularly noted the added value of guidance by a physical therapist as training volume could be adjusted in times of increased fatigue and decreased mood. Adding supervision to the home-based program might facilitate personalization of the physical exercise training program, which can improve adherence and motivation to the home-based program.

Patients in this study indicated that their informal caregivers were a major source of social support that motivated them to continue the rehabilitation program. In the patient with low adherence, there was no involvement of an informal caregiver. Among patients with cancer, social support is recognized as a positive determinant of adherence to a prehabilitation program [41]. Informal caregivers are a major source of social support and may influence patient’s physical activity adoption and maintenance by serving as role models and motivators [42,43]. A previous study reported that the supervision of healthcare professionals played an important role in the completion of prehabilitation in patients with cancer, as the patients needed to be pressured, monitored, and controlled [44]. Supervision as part of our intervention, along with increased social support at home, may have resulted in better adherence to the intervention. A tool to improve adherence could be the use of tele-monitoring [45]. In patients performing cardiac rehabilitation, technologies such as tele-monitoring can improve motivation and adherence, as coaching and encouragement are perceived as positive, and the supervision and adjustment of training intensity can help promote adherence through tele-monitoring while conducting their home-based physical exercise training sessions [45,46].

### 4.1. Strengths and Limitations

The present study provides detailed qualitative and quantitative data on the feasibility of a multimodal rehabilitation program during CHRT. This allowed patients to explain to themselves how, why, or what they thought, felt, and experienced at a certain time or during CHRT. This combination of qualitative and quantitative data provides deeper insights into real-world problems and aggregates patient experiences, preferences, and facilitators alongside quantitative data. However, there were also some limitations. First, program adherence was mainly assessed using the patient’s diary, potentially posing a risk of bias on reliability that might over- or underestimate actual training frequency, intensity, and duration despite regular contact with healthcare professionals. Patients undergoing CHRT experience disease-related and treatment-related impairments, for which a diary could be supportive. Second, although it was not the primary aim of the study, it was not possible to measure whether there was actually a significant improvement in physical performance due to the small sample size. Alternatively, demonstrating the feasibility of the program as well as preliminary effects of the training on relevant end points is a necessary first step to generate qualitative and detailed information regarding the feasibility and experiences before the initiation of randomized controlled trials.

### 4.2. Implications and Future Research

Rehabilitation programs during CHRT can improve overall health and lifestyle in multiple areas, such as physical exercise training, nutritional support, and smoking-cessation, especially with supervision of healthcare professionals [47,48]. In patients with NSCLC undergoing CHRT, low-intensity training during CHRT seems recommended with regard to feasibility. To possibly achieve a similar training volume as moderate- to high-intensity training, it is recommended to use the results of this rehabilitation program to design a supervised and personalized rehabilitation program during CHRT with a low-to-moderate training intensity and a longer training session duration. Improving treatment outcomes in a joint coalition with patients with cancer, supported by multimodal rehabilitation during CHRT, is an emerging therapeutic area. Future research should attempt to optimize the adherence of the exercise training intensity of rehabilitation during CHRT, e.g., through a combination of physically guided and/or tele-monitored supervision. Moreover, a larger prospective observational study should be designed to evaluate the effectiveness of multimodal rehabilitation on aerobic fitness and treatment outcomes (e.g., long-term health-related quality of life, complications, and physical (aerobic) fitness). These studies should take the effects of different interventions (e.g., supervised, partly supervised, home-based, tele-monitoring) into account.

## 5. Conclusions

Supervised and personalized rehabilitation in patients with stage III NSCLC undergoing CHRT seems feasible when the intensity of the physical exercise training program and nutritional advice are adjusted to the possibilities and preferences of the patients. Furthermore, a large variety existed in response to rehabilitation as some patients showed large improvements in preliminary effects on physical and nutritional outcome measures, whereas others showed no progression or deteriorated. It is therefore recommended to use the results of this proof-of-concept study to investigate the feasibility of a supervised and personalized rehabilitation program during CHRT with a low-to-moderate exercise intensity with the aim to prevent physical decline during CHRT.

## Figures and Tables

**Figure 1 cancers-14-02387-f001:**
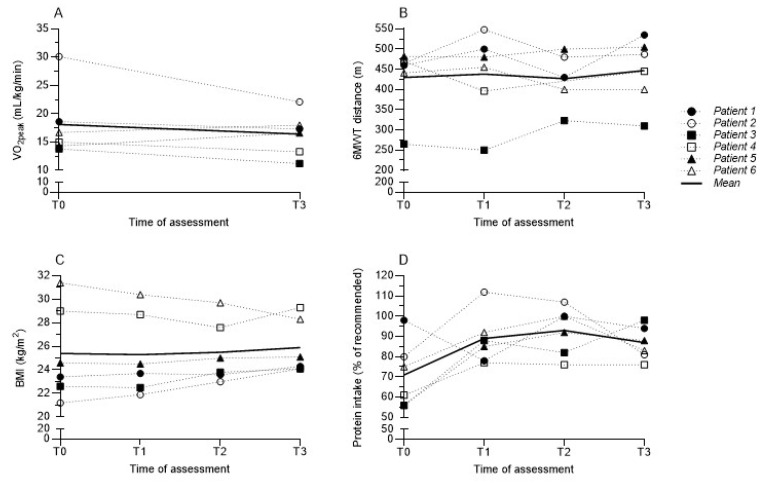
Preliminary changes in physical and nutritional assessments at T0, T1, T2, and T3. Graphs represent individual outcomes of the CPET (Absolute oxygen uptake at peak exercise) (**A**), 6MWT (**B**), BMI (**C**), and protein intake (**D**). The thick solid line represents mean values. Abbreviations: 6MWT = six-minute walk test; BMI = body mass index; CPET = cardiopulmonary exercise test; VO_2peak_ = oxygen uptake at peak exercise.

**Table 1 cancers-14-02387-t001:** Schedule of enrolment, interventions, and assessments for patients who underwent rehabilitation during chemoradiotherapy.

	Assessments	Appointment	T0	Appointment	T1	T2	T3
	Concurrent CHRT (cCHRT)	Week 0	Week 1	Week 2	Week 5	Week 13	Week 22
	Sequential CHRT (sCHRT)	Week 0	Week 1	Week 2	Week 5	Week 19	Week 28
ENROLMENT
	Informed consent						
	Informed about smoking						
CANCER TREATMENT:						
	Consultation with pulmonologist						
	Intake by case manager						
	Chemotherapy			Start		
	Radiotherapy		For cCHRT: start during CT; for sCHRT: start after CT
MULTIMODAL REHABILITATION DURING CHRT:	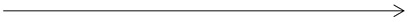
	Physical counseling ^a^						
	Dietary counseling ^b^						
	Case manager ^c^						
ASSESSMENTS:						
	CPET						
	6MWT						
	HGS						
	BMI						
	FFMI						
	Energy and protein intake						
	Pedometer						
FEASBILITY
	Adherence and dropouts						
	Smoking						
	0–10 VAS score for motivation						

Abbreviations: 6MWT = six-min. walking test; BMI = body mass index; CHRT = chemoradiotherapy; cCHRT = concurrent chemoradiotherapy; CPET = cardiopulmonary exercise test; CT = chemotherapy; FFMI = fat free mass index; HGS = handgrip strength; sCHRT = sequential chemoradiotherapy; VAS = visual analogue scale. ^a^: once every two weeks: supervision of the exercise program at the patient’s home or a visit during treatment with chemotherapy. ^b^: Telephone consultation every three weeks during chemotherapy and every week during radiotherapy. ^c^: Every three weeks by telephone.

**Table 2 cancers-14-02387-t002:** Blueprint of the physical exercise training program according to the i-CONTENT scale.

Patient Selection ^a^	Patients Aged ≥50 Years Diagnosed with Stage III NSCLC According to the 8th Edition of the TNM Guidelines Undergoing CHRT (Either Concurrent CHRT or Sequential CHRT)
**Type and dosage of the rehabilitation program during CHRT** **(F: Frequency, I: intensity, T: Time, T: Type)**	Aerobic exercises: F: 5 times/week 30 min, I: 6–20 Borg score 13–15, T: 30–60 min, T: Functional exercises involving large muscle groups (e.g., walking, cycling, climbing stairs, and swimming) Resistance exercises: F: 3 times/week, I: 6–20 Borg score 13–15, T: 3 × 15–20 repetitions, T: Peripheral resistance training of the large muscle groups of the lower and upper extremities using open and closed kinetic chain exercises (e.g., stair climbing, sit-to stand exercises, a Thera band, filled 0.5 L bottles) Breathing exercises: F: 2/day, I: highest tolerable intensity, T: 30 breaths, T: Inspiratory muscle training
**Qualified supervisor (if applicable)**	The physical exercise training program was carried out in the patient’s living environment, every two weeks supervised by a physical therapist specialized in oncology
**Type and timing of outcome assessment**	Type: feasibility of the multimodal rehabilitation program during CHRT was measured by the patient’s preferences and experiences, patient dropout, and adverse events during rehabilitation, adherence to the rehabilitation program, motivation, and problems concerning logistic planning Timing: before the start of CHRT (T0), between the first and second chemotherapy (T1), after the last session of radiotherapy (T2), three months after the last treatment (T3)
**Safety of the exercise program**	Patient dropout and adverse events to rehabilitation during CHRT were registered by the healthcare professionals during contact moments as part of usual care
**Adherence to the exercise program**	Adherence was monitored with a diary and weekly feedback from the patients. Successful exercise session adherence was defined as achieving >80% of the prescribed duration, intensity, and frequency of the training sessions during the physical exercise training program

Abbreviations: CHRT = chemoradiotherapy; i-CONTENT=international consensus on therapeutic exercise and training; NSCLC = non-small cell lung cancer. ^a^ Items of the i-CONTENT tool are presented in bold.

**Table 3 cancers-14-02387-t003:** Patient characteristics, treatment schedule, physical and nutritional parameters, and feasibility of rehabilitation during CHRT in patients with stage III NSCLC.

Variable	Patient 1				Patient 2				Patient 3			
Age	57				60				74			
Sex	Male				Male				Male			
Stage	IIIB				IIIB				IIIA			
Comorbidities	None				None				CABG (2017)			
Time of assessment	T0	T1	T2	T3	T0	T1	T2	T3	T0	T1	T2	T3
Treatment schedule												
Type of CHRT	Concurrent				Concurrent				Sequential			
Treatment time of CT	8 weeks				9 weeks				16 weeks			
Treatment time of RT	6 weeks				4 weeks				6 weeks			
Physical parameters												
CPET VO_2peak_ (mL/kg/min)	17.7	-	-	17.8	27.1	-	-	22.1	13.8	-	-	11.7
CPET VO_2_ at the VAT (mL/kg/min)	9.3	-	-	8.9	16.6	-	-	12.2	8.7	-	-	7.0
CPET RER_peak_	1.02	-	-	1.02	1.21	-	-	1.19	1.07	-	-	1.04
6MWT (m)	460	500	430	535	465	548	480	487	265	250	323	310
6MWT 6–20 Borg score	11	11	12	11	10	12	10	11	13	14	13	12
HGS dominant hand (kg)	46	48	48	52	37	NM	30	25	31	26	28	25
Nutritional parameters												
Body mass (kg)	74.0	75.3	74.8	77.1	62.9	66.3	69.5	72.9	63.0	62.6	66.4	67.3
BMI (kg/m^2^)	23.4	23.7	23.6	24.3	21.2	21.9	23.0	24.1	22.6	22.5	23.8	24.1
FFMI (kg/m^2^)	16.6	16.5	16.9	16.7	16.7	17.4	17.3	17.2	15.3	14.9	16.9	15.6
Energy intake (% of recommended)	93	113	90	85	95	116	102	99	53	92	90	99
Protein intake (% of recommended)	98	78	100	94	80	112	107	81	56	88	82	98
Feasibility												
Adherence to rehabilitation	100%				80%				48%			
6–20 Borg score during exercises	13	12	11	13	11	10	10	11	10	10	10	10
Smoking	yes	no	no	yes	yes	no	no	yes	yes	yes	yes	yes
0–10 VAS for motivation to perform rehabilitation	10	10	10	9	7	8	8	9	7	8	9	10

Abbreviations: 6MWT = six-minute walk test; BMI = body mass index; CABG = coronary artery bypass graft; CHRT = chemoradiotherapy; CPET = cardiopulmonary exercise test; CT = chemotherapy; FFMI = fat free mass index; HGS = handgrip strength; NSCLC = non-small cell lung cancer; RER_peak_ = respiratory exchange ratio at peak exercise; RT = radiotherapy; VAS = visual analogue scale; VAT = ventilatory anaerobic threshold; VO_2peak_ = oxygen uptake at peak exercise.

**Table 4 cancers-14-02387-t004:** Patient characteristics, treatment schedule, physical and nutritional parameters, and feasibility of rehabilitation during CHRT in patients with stage III NSCLC continued.

Variable	Patient 4				Patient 5				Patient 6			
Age	58				70				69			
Sex	Female				Male				Male			
Stage	IIIA				IIIA				IIIA			
Comorbidities	Depression (since 1995)				Osteopenia (since 2017)				RA (since 2010)			
Time of assessment	T0	T1	T2	T3	T0	T1	T2	T3	T0	T1	T2	T3
Treatment schedule												
Type of CHRT	Concurrent				Concurrent				Sequential			
Treatment time of CT	8 weeks				10 weeks				19 weeks			
Treatment time of RT	5 weeks				4 weeks				5 weeks			
Physical parameters												
CPET VO_2peak_ (mL/kg/min)	15.0	-	-	13.3	14.3	-	-	16.6	19.2	-	-	18.3
CPET VO_2_ at the VAT (mL/kg/min)	9.0	-	-	10.9	12.0	-	-	10.0	11.1	-	-	14.1
CPET RER_peak_	1.04	-	-	1.02	1.21	-	-	1.36	1.06	-	-	1.03
6MWT (m)	470	396	- ^a^	445	482	480	500	505	441	455	400	400
6MWT 6–20 Borg score	12	12	- ^a^	12	12	11	11	13	12	12	13	13
HGS dominant hand (kg)	29	30	- ^a^	32	30	38	38	35	26	28	25	29
Nutritional parameters												
Body mass (kg)	79.9	81.2	77.0	81.8	77.6	77.8	79.2	79.5	99.4	96.3	94.1	89.5
BMI (kg/m^2^)	29.0	28.7	27.6	29.3	24.6	24.5	25.0	25.1	31.4	30.4	29.7	28.3
FFMI (kg/m^2^)	17.2	17.4	- ^a^	17.4	17.8	18.5	18.3	18.9	19.2	17.8	17.5	17.6
Energy intake (% of recommended)	69	88	92	87	81	112	115	113	60	86	100	64
Protein intake (% of recommended)	61	77	76	76	56	85	92	88	75	92	100	83
Feasibility												
Adherence to rehabilitation	80%				100%				80%			
6–20 Borg score during exercises	11	12	10	12	11	11	11	11	13	12	10	12
Smoking	yes	no	no	Yes	No	No	No	No	No	No	No	No
0–10 VAS for motivation to perform rehabilitation	8	7	8	8	9	9	8	8	7	8	6	9

Abbreviations: 6MWT = six-minute walk test; BMI = body mass index; CHRT = chemoradiotherapy; CPET = cardiopulmonary exercise test; CT = chemotherapy; FFMI = fat free mass index; HGS = handgrip strength; NSCLC = non-small cell lung cancer; RA = rheumatoid arthritis; RER_peak_ = respiratory exchange ratio at peak exercise; RT = radiotherapy; VAS = visual analogue scale; VAT = ventilatory anaerobic threshold; VO_2peak_ = oxygen uptake at peak exercise. ^a^: not assessed due to COVID-19 restrictions.

## Data Availability

The data presented in this study are available in the article.

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
