# Peer review of "Feasibility of Rehabilitation during Chemoradiotherapy among Patients with Stage III Non-Small Cell Lung Cancer: A Proof-of-Concept Study"

_cancers, 2022, doi:10.3390/cancers14102387_

Round 1
Reviewer 1 Report
Dear authors,
I like to congratulate you for a very nice idea combining supervised and non-supervised rehabilitation programs designed taking patient´s advise in mind.
I have enjoyed reading the paper.
There is a minor comment to help improve the manuscript:
Abstract: Although this is an observational study in a short cohort of patients, it would be interesting to detail the recorded variables and the way those variables were analyzed in the abstract to give the reader a better idea of what they are going to find in the text.
No other questions or remarks to do but to expect a full prospective study following these exciting results.
Author Response
There is a minor comment to help improve the manuscript:
We would like to thank the reviewer for the positive feedback.
Abstract: Although this is an observational study in a short cohort of patients, it would be interesting to detail the recorded variables and the way those variables were analyzed in the abstract to give the reader a better idea of what they are going to find in the text.
- We agree with the reviewer that it is relevant to include information about the recorded variables and the way those variables were analyzed in the abstract. The following sentence has been added to the abstract: “The patient’s level of physical functioning (e.g., cardiopulmonary exercise test, six-minute walking test, handgrip strength, body mass index, fat free mass index, energy and protein intake) was evaluated in order to provide personalized advice regarding physical exercise training and nutrition.”
Reviewer 2 Report
General comments
This proof-of-concept study aimed at investigating the feasibility of a home-based multimodal program for rehabilitation, i.e., a partly supervised physical exercise training of moderate-intensity along with nutritional support, during chemoradiotherapy in patients with stage III non-small cell lung cancer (NSCLC).
Previous studies have shown that patients with lung cancer perceive physical activity as being important for their recovery during and after treatment. Nevertheless, most patients are insufficiently active, while evidence regarding the feasibility of a multimodal rehabilitation program during chemoradiotherapy, particularly in patients with stage III NSCLC, appears to be lacking.
Thus, this is a timely and interesting study, as revealing the feasibility of such a rehabilitation program and the preliminary effects of the exercise training are important steps for the generation of more specific information before the initiation of randomized controlled trials.
The following points should be addressed by the authors.
Specific comments
- Did the authors perform a sample size calculation for this proof-of-concept study (J Biopharm Stat. 2002 May;12(2):267-76)? Only seven patients invited to participate in such type of study within a period of 2 years appear to be a disproportionately small number, especially considering the different treatment schedules that followed and the participation of patients of both genders; please explain.
- Although the small number of patients (six) included in this study might not support a valid statistical testing, however a non-parametric test could be used to provide just a “clue” of the probability of making a Type II error.
Author Response
We would like to thank the reviewer for the positive notes and points for improvement.
The following points should be addressed by the authors.
Specific comments
Did the authors perform a sample size calculation for this proof-of-concept study (J Biopharm Stat. 2002 May;12(2):267-76)? Only seven patients invited to participate in such type of study within a period of 2 years appear to be a disproportionately small number, especially considering the different treatment schedules that followed and the participation of patients of both genders; please explain.
- We agree with the reviewer that the inclusion of patients requires explanation in the manuscript. No formal sample size calculation was performed for this study. Due to logistical difficulties and the impact of the COVID-19 pandemic, we were unfortunately limited in including patients for this rehabilitation program during chemoradiation. We added the following sentences to the results section to explain the problem with patient inclusion: “The time between diagnosis and inclusion of patients was short, partly due to additional diagnostic investigations. The consequence of the multiple healthcare organizations involved was a complexity in patient referrals and therefore patient inclusion for this study has been difficult. In addition, inclusion was made more difficult by the impact of the COVID-19 pandemic”
Although the small number of patients (six) included in this study might not support a valid statistical testing, however a non-parametric test could be used to provide just a “clue” of the probability of making a Type II error.
- Although preventing decline in physical fitness and nutritional status is already of great benefit in this patient population, we also considered presenting outcomes of a statistical test in the manuscript,. However for a statistical test we would have preferred a control group to be able to correctly determine the effect interpret in this context. Therefore, we have made choices because of the relevance of the information. As a result, we decided to present the detailed outcomes of each patient and graphs on the most important preliminary changes in physical and nutritional assessments.